# Associated Factors of Sarcopenia in Community-Dwelling Older Adults: A Systematic Review and Meta-Analysis

**DOI:** 10.3390/nu13124291

**Published:** 2021-11-27

**Authors:** Qianqian Gao, Kaiyan Hu, Chunjuan Yan, Bing Zhao, Fan Mei, Fei Chen, Li Zhao, Yi Shang, Yuxia Ma, Bin Ma

**Affiliations:** 1Evidence-Based Medicine Center, School of Basic Medical Sciences, Lanzhou University, Lanzhou 730000, China; gaoqq19@lzu.edu.cn (Q.G.); huky18@lzu.edu.cn (K.H.); lzzbing2019@163.com (B.Z.); 18821751725@163.com (F.M.); chenf_2019@163.com (F.C.); zhaoli545554617@163.com (L.Z.); 2Evidence-based Nursing Center, School of Nursing, Lanzhou University, Lanzhou 730000, China; yuxiama@lzu.edu.cn; 3School of Public Health, Gansu University of Traditional Chinese Medicine, Lanzhou 730000, China; ycj202111@163.com; 4Department of General Surgery, The Second Hospital, Lanzhou University, Lanzhou 730000, China; shangwkh6@163.com; 5Key Laboratory of Evidence-Based Medicine and Knowledge Translation of Gansu Province, Lanzhou 730000, China

**Keywords:** older adults, sarcopenia, associated factors, community, systematic review, meta-analysis

## Abstract

(1) Background: To review the associated factors of sarcopenia in community-dwelling older adults. (2) Methods: PubMed, Embase, Web of Science, and four Chinese electronic databases were searched for observational studies that reported the associated factors of sarcopenia from inception to August 2021. Two researchers independently selected the literature, evaluated their quality, and extracted relevant data. The pooled odds ratio (OR) and its 95% confidence interval (CI) were calculated for each associated factors of sarcopenia using random-effects/fixed-effects models. Publication bias was assessed using funnel plot and the Eggers test. We performed statistical analysis using Stata 15.0 software. (3) Results: A total of 68 studies comprising 98,502 cases were included. Sociodemographic associated factors of sarcopenia among community-dwelling older adults included age (OR = 1.12, 95% CI: 1.10–1.13), marital status (singled, divorced, or widowed) (OR = 1.57, 95% CI: 1.08–2.28), disability for activities of daily living (ADL) (OR = 1.49, 95% CI: 1.15–1.92), and underweight (OR = 3.78, 95% CI: 2.55–5.60). Behavioral associated factors included smoking (OR = 1.20, 95% CI: 1.10–1.21), physical inactivity (OR = 1.73, 95% CI: 1.48–2.01), malnutrition/malnutrition risk (OR = 2.99, 95% CI: 2.40–3.72), long (OR = 2.30, 95% CI: 1.37–3.86) and short (OR = 3.32, 95% CI: 1.86–5.93) sleeping time, and living alone (OR = 1.55, 95% CI: 1.00–2.40). Disease-related associated factors included diabetes (OR = 1.40, 95% CI: 1.18–1.66), cognitive impairment (OR = 1.62, 95% CI: 1.05–2.51), heart diseases (OR = 1.14, 95% CI: 1.00–1.30), respiratory diseases (OR = 1.22, 95% CI: 1.09–1.36), osteopenia/osteoporosis (OR = 2.73, 95% CI: 1.63–4.57), osteoarthritis (OR = 1.33, 95% CI: 1.23–1.44), depression (OR = 1.46, 95% CI: 1.17–1.83), falls (OR = 1.28, 95% CI: 1.14–1.44), anorexia (OR = 1.50, 95% CI: 1.14–1.96), and anemia (OR = 1.39, 95% CI: 1.06–1.82). However, it remained unknown whether gender (female: OR = 1.10, 95% CI: 0.80–1.51; male: OR = 1.50, 95% CI: 0.96–2.34), overweight/obesity (OR = 0.27, 95% CI: 0.17–0.44), drinking (OR = 0.92, 95% CI: 0.84–1.01), hypertension (OR = 0.98, 95% CI: 0.84–1.14), hyperlipidemia (OR = 1.14, 95% CI: 0.89–1.47), stroke (OR = 1.70, 95% CI: 0.69–4.17), cancer (OR = 0.88, 95% CI: 0.85–0.92), pain (OR = 1.08, 95% CI: 0.98–1.20), liver disease (OR = 0.88, 95% CI: 0.85–0.91), and kidney disease (OR = 2.52, 95% CI: 0.19–33.30) were associated with sarcopenia. (4) Conclusions: There are many sociodemographic, behavioral, and disease-related associated factors of sarcopenia in community-dwelling older adults. Our view provides evidence for the early identification of high-risk individuals and the development of relevant interventions to prevent sarcopenia in community-dwelling older adults.

## 1. Introduction

The global population is aging rapidly due to a combination of increased life expectancy and falling fertility. According to the World Population Prospects 2019 released by the United Nations (UN), it will reach to 9.8 billion by 2050, of which more than 1.5 billion will be aged over 65 years, accounting for 16% of the total population [1]. Therefore, the health of the elderly will have increasing importance for an aging society, worldwide.

Sarcopenia is a geriatric syndrome characterized by progressive and generalized loss of skeletal muscle mass and function [2]. It has recently become a focus of interest in the field of geriatric research and clinical practice. Over recent years, a number of international groups have published diagnostic criteria for sarcopenia, including European Working Group on Sarcopenia in Older People (EWGSOP) [3] and its revised version [4], International Working Group on Sarcopenia (IWGS) [5], the Foundation for the National Institutes of Health (FNIH) sarcopenia project [6], and Asian Working Group for Sarcopenia (AWGS) [7], and they all think that the criteria should include low muscle mass and/or muscle strength, plus low physical function. In addition, Malmstrom and Morley created a simple screening tool for sarcopenia in older adults known as SARC-F [8], the validity of which has been confirmed. Estimates for the prevalence of sarcopenia in older adults dwelling in the community range from 9.9 to 40.4% [9], depending on the different diagnostic criteria used. Multiple studies have demonstrated that sarcopenia is associated with a variety of adverse health outcomes, such as functional decline, disability, healthcare costs, falls, fractures, hospitalization, longer hospital stay, and mortality in elderly adults [10,11,12]. Sarcopenia was also significantly associated with poorer prognosis of cancers of the digestive system, head and neck cancer, lung cancer, urothelial cancer, hematological malignancy, breast cancer, and ovarian cancer [13]. Moreover, sarcopenia also significantly increased the risk of some diseases, such as dysphagia [14], cognitive impairment [15], hypertension [16], diabetes [17], osteoporosis [18], nonalcoholic fatty liver disease, [19] and depression [20]. Because the exact pathogenesis of sarcopenia remains unclear, there are currently no targeted drug treatments [2]. Therefore, the identification of potentially modifiable associated factors for sarcopenia is of key importance for the early implementation of effective interventions to reduce sarcopenia and the adverse events with which it is associated.

The causes of sarcopenia are known to be multifactorial, including aging itself, sociodemographic factors, lifestyle, and a variety of health conditions [21]. However, the associated factors of sarcopenia varied in previous studies, and sometimes the results were inconsistent and controversial [22,23,24,25,26]. Currently, there is no consensus on the associated factors for sarcopenia. We all know that the prevention of sarcopenia in the community is not only of particular significance for improving the health of elderly residents; it will also be of decisive importance for promoting the health of the whole of society.

Therefore, we conducted a systematic review and meta-analysis to review the associated factors of sarcopenia among community-dwelling older adults, to allow early identification of those associated factors so that appropriate intervention measures can be taken in the future to reduce the occurrence of sarcopenia and its adverse events, thereby improving the health level of the elderly.

## 2. Materials and Methods

Our view was conducted in accordance with the Preferred Reporting Items for Systematic Reviews and Meta-Analyses (PRISMA) statement [27].

### 2.1. Search Strategy

Two researchers independently searched the following English and Chinese electronic databases from inception to August 2021: PubMed, Embase, Web of Science, China Biological Medicine (CBM) databases, Wan Fang, Chinese Scientific Journals Full-Text, and China National Knowledge Infrastructure (CNKI). The search terms were: aged, elderly, older people, older adults, sarcopenia, sarcopenic, muscle mass, muscle strength, muscular atrophy, gait speed, grip strength, risk factors, influence factors, associated factors, precipitating factors, and contributing factors. The references of all literature included in the review and previous relevant systematic reviews were also checked for any additional studies. Detailed search strategy is presented in Appendix A.

### 2.2. Eligibility Criteria

Inclusion criteria for the review were as follows: (1) participants: older adults ≥60 years old in community, without restrictions to gender, race, or nationality; (2) exposure: at least one sociodemographic, behavioral or disease-related associated factor of sarcopenia reported; (3) outcomes: sarcopenia, with unrestricted diagnostic criteria or threshold values; (4) study design: cohort, case-control, and cross-sectional studies; (5) articles in Chinese and English.

Exclusion criteria were as follows: (1) patients with specific serious diseases; (2) patients in hospital and nursing homes; (3) not reporting a clear diagnostic criterion for sarcopenia; (4) duplicates.

### 2.3. Study Selection

Two researchers independently selected the literature. Duplicates were first removed through the use of Endnote X9 software, after which the titles and abstracts were screened. Finally, the full text of each article was read to identify all studies for inclusion. The reasons that each article was excluded in the second and final steps were recorded. Any discrepancies were resolved by a third researcher.

### 2.4. Data Extraction

Two researchers independently extracted the data of the included studies using Microsoft Excel 2019. The following details were extracted: (1) the year of publication, the name of first author and study design; (2) region and setting in which the study was conducted; (3) total sample size, proportion of female participants and mean age; (4) diagnostic criteria of sarcopenia; (6) associated factors. Any inconsistencies were resolved by a third researcher. If it was necessary to confirm the data, we contacted the corresponding author of the study on two occasions within a month.

### 2.5. Quality Assessment

Two researchers independently assessed the quality of the included studies. For cross-sectional studies, we used the Joanna Briggs Institute Critical Appraisal Checklist for Analytical Cross-Sectional Studies [28]. The checklist consisted of eight items, which were evaluated “not applicable”, “unclear”, “no”, or “yes” for each study. The total number of “yes” answers was recorded for each study. Greater numbers of “yes” answers were representative of higher quality studies. For case-control and cohort studies, we used the Newcastle–Ottawa Scale (NOS) [29], consisting of three broad categories of eight items, with a maximum score of nine points. Studies with a NOS score < 5 are considered to have high risk of bias; 5 to 7, a moderate risk; and >7, a low risk. Any inconsistencies were resolved by a third researcher.

### 2.6. Statistical Analysis

We conducted meta-analysis using Stata version 15 software, with *p*-values < 0.05 being considered statistically significant. For each associated factor, the odds ratio (OR) and its 95% confidence interval (CI) were extracted. Pooled ORs and their 95% CIs were calculated for each associated factor. We assessed the statistical heterogeneity among the studies using a chi-square test and the degree quantified using the I^2^ statistic. Substantial heterogeneity was indicated where *p* ≤ 0.10 and I^2^ ≥ 50%. A fixed-effects model was used in the absence of substantial heterogeneity; otherwise, a random-effects model was employed.

We conducted subgroup analysis by the diagnostic criteria of sarcopenia (muscle mass, alone; standards proposed by some international groups, SARC-F), whether to adjust for confounding factors (adjusted, unadjusted) and geographical region (Asia, Europe, South America, North America, and multicenter). We assessed the publication bias using funnel plot and the Eggers test (*p* < 0.05). We also performed the trim-and-fill analysis to assess the effects of publication bias.

## 3. Results

### 3.1. Search Results

We identified 33,572 articles in the initial search of the literature and additional records through other sources, of which 29,966 remained after removal of duplicate articles. After we screened the titles and abstracts, there were 111 studies selected for full-text evaluation. Finally, 43 articles were excluded, and the rest of the 68 articles [22,23,24,25,26,30,31,32,33,34,35,36,37,38,39,40,41,42,43,44,45,46,47,48,49,50,51,52,53,54,55,56,57,58,59,60,61,62,63,64,65,66,67,68,69,70,71,72,73,74,75,76,77,78,79,80,81,82,83,84,85,86,87,88,89,90,91,92] met the eligibility criteria. The detailed selection process is presented in Figure 1.

### 3.2. Characteristics of Included Studies and Participants

Of the studies included in the review, six were prospective cohort studies [38,44,45,46,71,92], and 62 were cross-sectional. Only one study [53] published prior to 2010. The total number of participants finally included in the review was 98,502, with samples ranging in size from 120 [32] to 18,363 [26] patients. In the 58 studies, more than 50% of the participants were women. The majority of studies were conducted in Asia (*n* = 45), 14 in South America [24,31,32,43,55,56,57,60,68,69,78,83,88,91], six in Europe [46,48,50,70,80,88], one in North America [92], and two in multicenter. The EWGSOP (*n* = 28) and AWGS (*n* = 24) criteria were the most commonly used in the studies included in the review. SARC-F was used to define sarcopenia in three studies [64,70,86], while the IWGS criteria were used in another study [55]. One study [22] applied more than one diagnostic criterion to define sarcopenia, including EWGSOP, AWGS, IWGS, and FNIH. Finally, the remaining 11 studies [25,32,35,53,56,60,67,77,78,89,91] diagnosed sarcopenia using low muscle mass, alone. The detailed characteristics are summarized in Appendix A.

### 3.3. Quality Assessment of the Included Studies

The Joanna Briggs Institute Critical Appraisal Checklist was used to assess the quality of the 62 cross-sectional studies. Specifically, 7 studies fulfilled six items, and 23 studies fulfilled seven items, while the remaining 32 satisfied eight items. The Newcastle–Ottawa scale was used for the six cohort studies. All scored 9 and were considered to be of high quality. The quality assessment of cross-sectional and cohort studies is shown in Figure 2 and Figure 3, respectively.

### 3.4. Associated Factors of Sarcopenia in Community-Dwelling Older Adults

#### 3.4.1. Sociodemographic Factors

Of the studies included in the review, six sociodemographic factors were assessed, including age, gender, level of education, marital status, body mass index (BMI), and ADL disability. Meta-analyses demonstrated that age (34 studies, OR = 1.12, 95% CI: 1.10–1.13, Appendix A), marital status (singled, divorced, or widowed) (7 studies, OR = 1.57, 95% CI: 1.08–2.28, Appendix A), low BMI (underweight) (14 studies, OR = 3.78, 95% CI: 2.55–5.60, Appendix A), and ADL disability (7 studies, OR = 1.49, 95% CI: 1.15–1.92, Appendix A) were independent associated factors of sarcopenia. (Table 1). However, female (21 studies, OR = 1.10, 95% CI: 0.80–1.51, Appendix A), male (11 studies, OR = 1.50, 95% CI: 0.96–2.34, Appendix A), higher level of education (11 studies, OR = 0.95, 95% CI: 0.92–0.98, Appendix A), and high BMI (overweight/obesity) (12 studies, OR = 0.27, 95% CI: 0.17–0.44, Appendix A) were not associated with sarcopenia. (Table 2).

#### 3.4.2. Behavioral Factors

Six behavioral factors were assessed in the studies included in the review, including smoking, drinking, malnutrition/malnutrition risk, sleeping time, living alone, and physical activity. Meta-analyses showed that smoking (29 studies, OR = 1.20, 95% CI: 1.10–1.21, Appendix A), malnutrition/malnutrition risk (10 studies, OR = 2.99, 95% CI: 2.40–3.72, Appendix A), long (2 studies, OR = 2.30, 95% CI: 1.37–3.86, Appendix A) and short (2 studies, OR = 3.32, 95% CI: 1.86–5.93, Appendix A) sleeping time, living alone (5 studies, OR = 1.55, 95% CI: 1.00–2.40, Appendix A), and physical inactivity (18 studies, OR = 1.73, 95% CI: 1.48–2.01, Appendix A) were independent associated factors of sarcopenia. (Table 1). However, drinking (21 studies, OR = 0.92, 95% CI: 0.84–1.01, Appendix A) was not associated with sarcopenia. (Table 2).

#### 3.4.3. Disease-Related Factors

There were 17 disease-related factors were assessed in the included studies, including diabetes, hypertension, cognitive impairment, hyperlipidemia, heart diseases, stroke, respiratory diseases, cancer, osteopenia/osteoporosis, osteoarthritis, depression, fall, pain, anorexia, anemia, liver disease, and kidney disease. Meta-analyses showed that diabetes (19 studies, OR = 1.40, 95% CI: 1.18–1.66, Appendix A), cognitive impairment (6 studies, OR = 1.62, 95% CI: 1.05–2.51, Appendix A), heart diseases (5 studies, OR = 1.14, 95% CI: 1.00–1.30, Appendix A), respiratory diseases (7 studies, OR = 1.22, 95% CI: 1.09–1.36, Appendix A), osteopenia/osteoporosis (6 studies, OR = 2.73, 95% CI: 1.63–4.57, Appendix A), osteoarthritis (6 studies, OR = 1.33, 95% CI: 1.23–1.44, Appendix A), depression (11 studies, OR = 1.46, 95% CI: 1.17–1.83, Appendix A), fall (9 studies, OR = 1.28, 95% CI: 1.14–1.44, Appendix A), anorexia (2 studies, OR = 1.50, 95% CI: 1.14–1.96, Appendix A), and anemia (2 studies, OR = 1.39, 95% CI: 1.06–1.82, Appendix A) were independent associated factors of sarcopenia. (Table 1). However, hypertension (13 studies, OR = 0.98, 95% CI: 0.84–1.14, Appendix A), hyperlipidemia (5 studies, OR = 1.14, 95% CI: 0.89–1.47, Appendix A), stroke (4 studies, OR = 1.70, 95% CI: 0.69–4.17, Appendix A), cancer (5 studies, OR = 0.88, 95% CI: 0.85–0.92, Appendix A), pain (3 studies, OR = 1.08, 95% CI: 0.98–1.20, Appendix A), liver disease (3 studies, OR = 0.88, 95% CI: 0.85–0.91, Appendix A), and kidney disease (2 studies, OR = 2.52, 95% CI: 0.19–33.30, Appendix A) were not associated with sarcopenia. (Table 2).

### 3.5. Subgroup Analysis

We conducted subgroup analysis to discuss potential causes of heterogeneity using the diagnostic criteria of sarcopenia, whether to adjust for confounding factors and geographical region for those factors with high heterogeneity. The details were presented in Appendix A.

### 3.6. Publication Bias

For those associated factors reported more than in ten studies, we assessed the publication bias. There was evidence of publication bias according to the Egger test and funnel plots for the following associated factors: male (*p* = 0.04, Appendix A), overweight/obesity (*p* = 0.00, Appendix A), underweight (*p* = 0.00, Appendix A), high level of education (*p* = 0.00, Appendix A), smoking (*p* = 0.00, Appendix A), physical inactivity (*p* = 0.01, Appendix A), and hypertension (*p* = 0.03, Appendix A); but for depression (*p* = 0.01, Appendix A), age (*p* = 0.177, Appendix A), female (*p* = 0.972, Appendix A), malnutrition/malnutrition risk (*p* = 0.512, Appendix A), drinking (*p* = 0.32, Appendix A), and diabetes (*p* = 0.236, Appendix A), there was none. (Table 1 and Table 2). We conducted trim-and-fill analysis for those factors with publication bias and only found there were missing studies, and the overall effect size changed in the following associated factors: underweight (8 studies, OR = 1.82, 95% CI: 0.98–3.36), smoking (15 studies, OR = 1.12, 95% CI: 0.96–1.31), and physical inactivity (7 studies, OR = 1.45, 95% CI: 1.17–1.79), indicating that publication bias may affect these results.

## 4. Discussion

At present, most of the published meta-analyses related to sarcopenia focuses on prevalence, diagnosis, adverse health-related outcomes, and interventions, and little research has been done on its prevention. Our review was a first systematic review and meta-analysis about prevention to review the associated sociodemographic-, behavioral-, and disease-related factors of sarcopenia in community-dwelling older adults. We aimed to early identify those associated factors so that appropriate interventions can be taken to reduce the occurrence of sarcopenia. From the data in 68 studies, we found that the sociodemographic associated factors included age, marital status (singled, divorced, or widowed), ADL disability, and low BMI (underweight). Behavioral-associated factors included smoking, physical inactivity, malnutrition/malnutrition risk, both long and short sleeping time, and living alone. Disease-related associated factors included diabetes, cognitive impairment, heart diseases, respiratory diseases, osteopenia/osteoporosis, osteoarthritis, depression, falls, anorexia, and anemia. However, it remained unknown whether gender, drinking, overweight/obesity, hypertension, hyperlipidemia, stroke, cancer, anemia, liver disease, and kidney disease were independent associated factors of sarcopenia.

### 4.1. Sociodemographic Associated Factors of Sarcopenia in Community-Dwelling Older Adults

Our view found that age was an associated factor of sarcopenia in community-dwelling older adults. On one hand, it may be related to skeletal muscle mass and strength beginning to decline after the age of 30, the rate of which increases markedly over 60 years of age [93]. On the other hand, with increased age, muscle fibers gradually experience fibrosis or are replaced with adipose tissue, with oxidative damage that is exacerbated because the capability for antioxidant action declines. The type of muscle fibers change, with the ratio of type II/I fibers decreasing, the absolute mass of both declining, and so causing the mass and strength of the skeletal muscle to decrease [94]. As a result, elderly adults are at greater risk of sarcopenia as they age. ADL disability is associated with reduced motor capability and grip strength [65], the principal screening indicator for muscle degeneration. Individuals with ADL disability are at greater risk of sarcopenia than those with normal motor function. Therefore, individuals with daily living disabilities should be encouraged to perform passive physical activities to enhance muscle strength and function and prevent or reverse muscular atrophy. Furthermore, we found that a high level of education represented a protective factor for sarcopenia, possibly contributing to a healthier lifestyle, including superior nutrition and more intensive leisure time for physical activity over their life span, so this may be related to increased muscle mass and overall health status [48]. Additionally, marital status (single, divorced, or widowed) was an independent predictor of sarcopenia. Hence, health education should be strengthened for older people with relatively low levels of education, especially those that are single, divorced, or widowed. We also found that lower BMI (underweight) was an associated factor of sarcopenia, whereas higher BMI (overweight/obese) was not. Older adults that are underweight are therefore at increased risk of developing sarcopenia because they have insufficient protein intake and are more likely to suffer from malnutrition, while individuals with higher fat mass may have a higher protein intake, which is protective against sarcopenia [95]. Thus, a high BMI may act as a protective buffer to offset the loss of muscle performance in elderly adults. This phenomenon is known as the obesity paradox and has also been observed in patients with chronic obstructive pulmonary disease [96], chronic heart failure [97], and chronic kidney disease [98]. Moreover, we found that gender was not associated with increased risk of sarcopenia. This finding is consistent with a previous meta-analysis [99] that confirmed that the pooled prevalence of sarcopenia in both men and women was 10% in community-dwelling older adults.

### 4.2. Behavioral-Associated Factors of Sarcopenia in Community-Dwelling Older Adults

We found that smoking, both short and long sleeping time, physical inactivity, malnutrition/malnutrition risk, and living alone were independent factors that contribute to the development of sarcopenia. Previous published two systematic reviews also indicated cigarette smoking [100] and sleep duration [101] had a negative effect on sarcopenia. In addition, a meta-analysis [102] also discussed the association between sarcopenia and physical activity and found the beneficial influence of physical activity, in general, for the prevention of sarcopenia. Smoking can increase muscle fatigue, leading to protein catabolism disorders, thus reducing both muscle mass and function. Appropriate levels of sleep are critical to restoration and rejuvenation processes, while a duration that is too long or too short can dysregulate metabolic function, resulting in reduced muscle mass and physical function [101]. Thus, it is recommended that the elderly in the community quit smoking as soon as possible and maintain reasonable sleep patterns to reduce the risk of sarcopenia. Physical inactivity and malnutrition are two important causes of sarcopenia [2]. Physical activity is helpful for the recovery of mitochondrial metabolic function and reduced expression of catabolic genes, thereby increasing the synthesis of muscle protein. In addition, resistance exercise has been identified as an important strategy for the prevention of muscle atrophy, as it directly stimulates muscle hypertrophy and increases muscle strength. Furthermore, a number of nutritional factors, such as protein, vitamin D, and calcium intake play important roles in maintaining muscle mass and, consequently, reinforcing muscle strength and physical performance. As a result, the risk of sarcopenia increases in elderly adults when they do not undertake sufficient physical activity or have adequate nutrition. Beaudart et al. confirmed the effectiveness of exercise training and nutritional supplementation for the prevention and reversal of sarcopenia [103]. Therefore, for those elderly people who are not physically active or suffer from malnutrition, community healthcare workers should formulate appropriate exercise and nutritional programs depending on the specific health requirements of individuals. Additionally, living alone was also related to the occurrence of sarcopenia, and more attention should be paid to the elderly who live alone. Drinking was not found to be an associated factor for sarcopenia. However, due to the limitations in the original studies included in the review, we were unable to quantitatively analyze the relationship between alcohol intake and sarcopenia. Excessive alcohol consumption impairs skeletal muscle protein synthesis and the exposure of muscle tissue to ethanol results in autophagy, leading to sarcopenia [104]. Therefore, future prospective studies with large sample sizes should be conducted to further explore the relationship between consumption of alcohol and risk of sarcopenia.

### 4.3. Disease-Related Associated Factors of Sarcopenia in Community-Dwelling Older Adults

We found that diabetes, cognitive impairment, heart diseases, osteopenia/osteoporosis, osteoarthritis, respiratory diseases, depression, falls, anorexia, and anemia were independent factors contributing to the development of sarcopenia. Diabetes can accelerate the loss of muscle mass and strength due principally to insulin resistance, levels of inflammatory cytokines, and associated changes in endocrine function [30]. The specific mechanisms relating cognitive impairment and sarcopenia has not yet been fully illustrated. It may be related to a possible shared pathophysiology related to inflammatory markers and the hormonal pathway between them [15]. Excessive breakdown of muscle proteins and higher levels of inflammatory markers have been associated with sarcopenia in patients with heart disease [105]. Patients with osteoporosis exhibit low bone density, which can lead to low muscle mass, muscle strength, and reduced physical function, thus leading to a higher risk of sarcopenia. Osteoarthritis patients may suffer from joint pain and stiffness, resulting in limited physical activity that inevitably leads to muscle atrophy and loss of function. Respiratory diseases reduce physical activity due to exercise intolerance because of impaired lung function, limited gas exchange, and other factors, which are major causes of loss of muscle mass and dysfunction in the elderly. The principal causes of sarcopenia in depressed elderly adults are reduced physical activity, an irregular diet, and reduced nutrient intake [81]. Likewise, nutrient intake in anorexic individuals is also insufficient, resulting in sarcopenia. Elderly adults with a history of falls have impaired muscle function, and long-term repeated falls can further lead to a decline in muscle mass and strength. Therefore, our findings suggest that the management of such chronic diseases plays an important role in the prevention of sarcopenia among elderly adults dwelling within the community. In addition, previous systematic reviews have demonstrated that sarcopenia can significantly predict the development of diabetes [17], depression [20], and falls [106]. This means that there is a bidirectional relationship with sarcopenia. This evidence reinforces the conclusion that the prevention of sarcopenia is of critical importance. Furthermore, we found that hypertension, cognitive impairment, hyperlipidemia, heart disease, stroke, cancer, pain, anemia, and liver disease were not associated with increased risk of sarcopenia. However, only a handful of studies reported on those disease-specific associated factors, except for hypertension. Therefore, additional studies should be conducted in the future to explore the relationship between these disease-related associated factors and sarcopenia.

### 4.4. Implications of the Review for Future Clinical Practice

We concluded that there were many sociodemographic-, behavioral-, and disease-related associated factors of sarcopenia in community-dwelling older adults. Since some behavioral factors related to sarcopenia can be changed, community medical workers should do a good job of health education, including quitting smoking as soon as possible, maintaining appropriate sleeping time, and increasing nutritional supplements and regular physical activity to reduce the occurrence of sarcopenia. In terms of the sociodemographic factors, community health workers should pay special attention to the elderly, singled, divorced, widowed, and those with underweight and ADL disability, and screening for sarcopenia should be conducted as early as possible. For those disease-related factors, including diabetes, cognitive impairment, heart diseases, respiratory diseases, osteopenia/osteoporosis, osteoarthritis, depression, falls, anorexia, and anemia, community health workers should attach great importance to the management of diseases and guide their patients to take medicine regularly to reduce the occurrence of sarcopenia. On the other hand, routine screening of sarcopenia also should be performed so that timely interventions can be taken to reduce the risk of its adverse events.

Currently, the pathogenesis of sarcopenia is still unclear, so there are no approved medications for its treatment [2]. The most commonly proposed treatments are mainly exercise, nutrition, and both combinations, and they have been proven effectively in several studies. Luo et al. [107] discovered nutritional supplementation may have the positive effect on sarcopenia elderly for muscle strength, muscle mass, and physical performance. Vlietstra et al. [108] demonstrated exercise interventions significantly improved muscle strength, balance, and muscle mass. Furthermore, some meta-analyses [109,110,111] confirmed the combination of exercise and nutrition also have beneficial effects on physical performance and muscle strength in older adults with sarcopenia. From what has been discussed above, community health workers should early identify high-risk individuals and develop appropriate intervention measures to reduce the occurrence of sarcopenia and its adverse events, thereby improving the health level of the elderly.

### 4.5. Strengths and Limitations of The Review

Firstly, this was the first systematic review and meta-analysis to focus on the associated factors of sarcopenia in community-dwelling older adults. Secondly, an extensive literature search was conducted, with research papers comprehensively screened, in addition to other relevant literature, to reduce the likelihood as far as possible of missing any studies. Thirdly, most of the included studies were high-quality studies with reliable results.

However, our view also had some limitations. Firstly, a number of associated factors were reported in only a few studies, and so we were unable to establish a clear conclusion between them and sarcopenia by meta-analysis. Secondly, we found that some of the associated factors showed high heterogeneity, which may be related to the study geographic regions, the diagnostic criteria for sarcopenia, and whether or not confounders were adjusted and the amount of adjustment. Therefore, we used a random-effects model instead of a fixed-effects model for the quantitative analysis. In addition, we also conducted subgroup analysis for the above factors. Although the heterogeneity of some factors was reduced, reliable conclusions could not be drawn due to the small number of studies included in some subgroups. Nevertheless, heterogeneity is often inevitable in meta-analyses of observational studies, and it does not necessarily nullify the meta-analysis results [112]. Thirdly, the majority of studies included in the present review were of cross-sectional design and could not demonstrate causal relationships between the analyzed associated factors and sarcopenia, so it required future cohort studies to clarify. Finally, only those published in English and Chinese were included, which may have contributed to language bias. All of these limitations should be considered when interpreting the results.

## 5. Conclusions

In conclusion, age, marital status (singled, divorced, or widowed), ADL disability, underweight, smoking, physical inactivity, malnutrition/malnutrition risk, long or short sleeping time, living alone, diabetes, cognitive impairment, heart diseases, respiratory diseases, osteopenia/osteoporosis, osteoarthritis, depression, falls, anorexia, and anemia are associated factors of sarcopenia in community-dwelling older adults. Our findings provide evidence for the early identification of high-risk individuals and the development of relevant interventions to prevent sarcopenia in the elderly community. However, it remains unclear whether gender, overweight/obesity, drinking, hypertension, hyperlipidemia, stroke, cancer, pain, liver disease, and kidney disease are associated with sarcopenia. A greater number of multicenter and prospective cohort studies with large samples are required to further explore between these factors and sarcopenia in the future.

## Figures and Tables

**Figure 1 nutrients-13-04291-f001:**
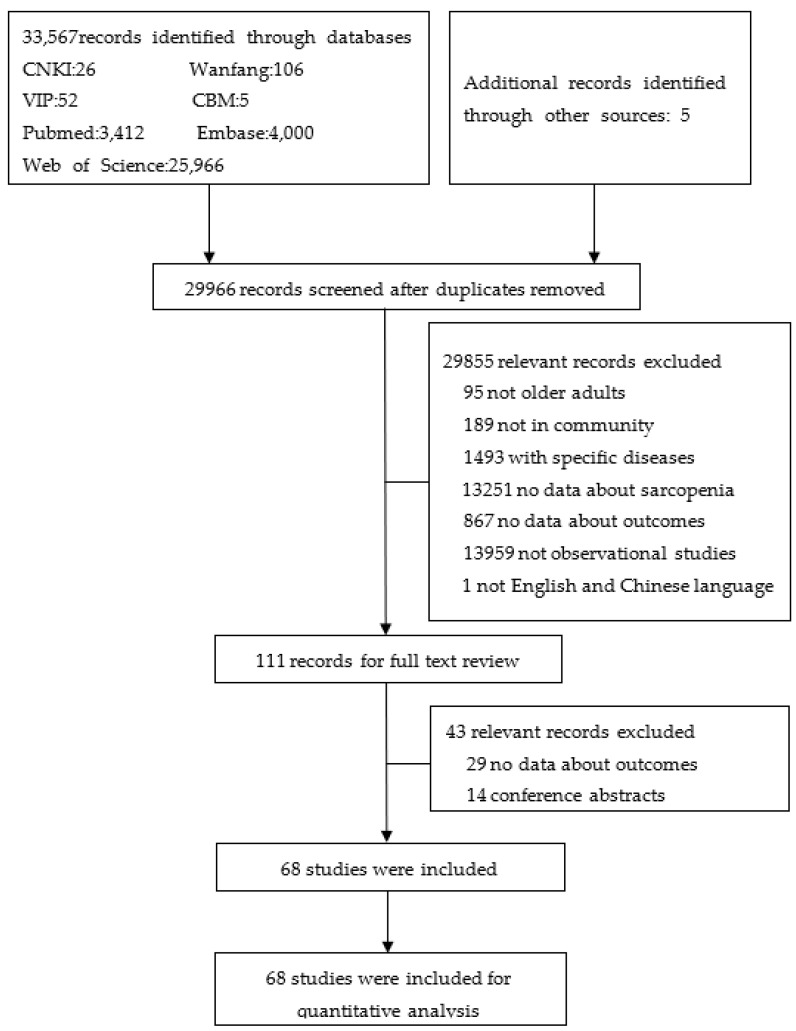
PRISMA flowchart.

**Figure 2 nutrients-13-04291-f002:**
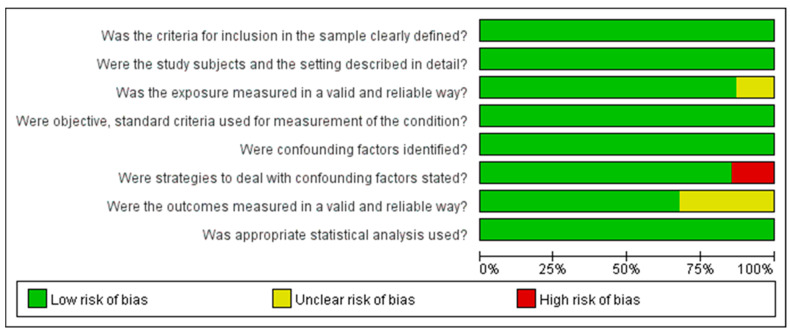
Quality assessment of the included cross-sectional studies.

**Figure 3 nutrients-13-04291-f003:**
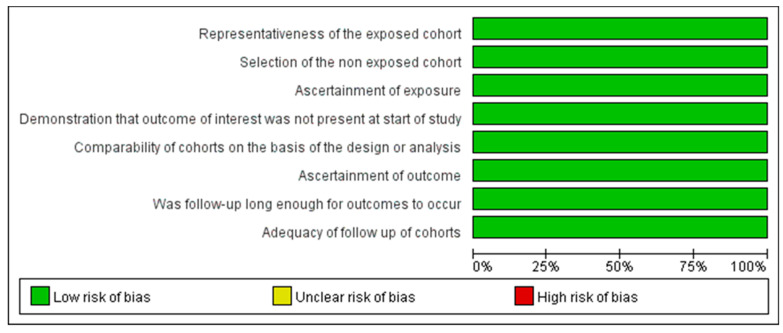
Quality assessment of the included cohort studies.

**Table 1 nutrients-13-04291-t001:** Pooled ORs and 95% confidence interval for associated factors of sarcopenia in community-dwelling older adults.

Associated Factors		Number of Studies	Heterogeneity		OR (95% CI)
			I^2^	*p*	
Sociodemographic factors	BMI Under weight	14	78.1	<0.001	3.78 (2.55, 5.60)
	Marital status	7	71.2	<0.001	1.57 (1.08, 2.28)
	ADL disability	7	84.7	<0.001	1.49 (1.15, 1.92)
	Age (years)	34	80.7	<0.001	1.12 (1.10, 1.13)
Behavioral factors	Sleeping time < 6 h	2	0.0	0.473	3.32 (1.86, 5.93)
	Malnutrition/malnutrition risk	10	46.7	0.029	2.99 (2.40, 3.72)
	Sleeping time ≥ 8 h	2	14.7	0.279	2.30 (1.37, 3.86)
	Physical inactivity	18	65.2	<0.001	1.73 (1.48, 2.01)
	Living alone	5	36.0	0.167	1.55 (1.00, 2.40)
	Smoking	29	49.5	<0.001	1.20 (1.10, 1.31)
Disease-related factors	Osteopenia/osteoporosis	6	75.2	<0.001	2.73 (1.63, 4.57)
	Cognitive impairment	6	69.9	0.001	1.62 (1.05, 2.51)
	Anorexia	2	0.0	0.425	1.50 (1.14, 1.96)
	Depression	11	69.4	<0.001	1.46 (1.17, 1.83)
	Diabetes	19	56.7	<0.001	1.40 (1.18, 1.66)
	Anemia	2	8.3	0.351	1.39 (1.06, 1.82)
	Osteoarthritis	6	32.7	0.167	1.33 (1.23, 1.44)
	Fall	9	0.0	0.461	1.28 (1.14, 1.44)
	Respiratory diseases	7	0.0	0.757	1.22 (1.09, 1.36)
	Heart diseases	5	0.0	0.966	1.14 (1.00, 1.30)

Notes: OR: odds ratio; ADL: activities of daily living; BMI: Body mass index; -: none.

**Table 2 nutrients-13-04291-t002:** Pooled ORs and 95% confidence interval for not associated factors of sarcopenia in community-dwelling older adults.

Not Associated Factors		Number of Studies	Heterogeneity		OR (95% CI)
			I^2^	*p*	
Sociodemographic factors	Male	11	86.4	<0.001	1.50 (0.96, 2.34)
	Female	21	86.1	<0.001	1.10 (0.80, 1.51)
	High level of education	11	60.8	<0.001	0.95 (0.92, 0.98)
	BMI Overweight/obesity	12	93.5	<0.001	0.27 (0.17, 0.44)
Behavioral factors	Drinking	21	48.3	0.001	0.92 (0.84, 1.01)
Disease-related factors	Kidney disease	2	78.2	0.032	2.52 (0.19, 33.30)
	Stoke	4	81.8	0.001	1.70 (0.69, 4.17)
	Hyperlipidemia	5	29.5	0.214	1.14 (0.89, 1.47)
	Pain	3	12.7	0.332	1.08 (0.98, 1.20)
	Hypertension	13	52.4	0.009	0.98 (0.84, 1.14)
	Cancer	5	0.0	0.542	0.88 (0.85, 0.92)
	Liver disease	3	26.6	0.256	0.88 (0.85, 0.91)

## Data Availability

Data sharing not applicable.

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
