# Peer review of "Associated Factors of Sarcopenia in Community-Dwelling Older Adults: A Systematic Review and Meta-Analysis"

_nutrients, 2021, doi:10.3390/nu13124291_

Round 1
Reviewer 1 Report
1. The authors list important articles in the manuscript, without their description, e.g.:
a) “Relationship between sarcopenia and physical activity in older people: a systematic review and meta-analysis”
b) “Health Outcomes of Sarcopenia: A Systematic Review and Meta-Analysis”
2. The authors do not cite and describe important articles, e.g.
a) “Interventions for Treating Sarcopenia: A Systematic Review and Meta-Analysis of Randomized Controlled Studies”
b) „The Role of Muscle Mass Gain Following Protein Supplementation Plus Exercise Therapy in Older Adults with Sarcopenia and Frailty Risks: A Systematic Review and Meta-Regression Analysis of Randomized Trials”
c) “Exercise interventions in healthy older adults with sarcopenia: A systematic review and meta-analysis”
d) “Sarcopenia and its association with falls and fractures in older adults: A systematic review and meta-analysis”
e) “Diagnosis, prevalence, and clinical impact of sarcopenia in COPD: a systematic review and meta-analysis”
The authors should review the link below and include the most important meta-analyzes in the manuscript. Authors should describe what distinguishes their article from other meta-analyzes published so far. This is the basis of my review. The text must contain missing articles (meta-analyzes) and a description of why your article is new.
https://pubmed.ncbi.nlm.nih.gov/?term=sarcopenia%20meta-analysis
3. The abstract should contain the most important results of the meta-analysis performed.
4. I would suggest a more detailed description of the early identification of high-risk individuals and the development of relevant interventions to prevent sarcopenia in the elderly community. The authors have discussed this aspect too generally.
5. For the sake of clarity, I would sort the predictors in the tables according to their increasing importance (based on the obtained results).
Author Response
Dear reviewer:
Thanks very much to you for giving us the opportunity to revise the review. According to your comments, we have made corresponding modifications in the review. Because the overall repetition rate of our review exceeded the 4% required by the journal, we tried to decrease repetition. In the process of decreasing repetition, we made a lot of small modifications while keeping the original contents unchanged. Using the function of track changes may make our review look particularly messy, so I marked the contents you suggested to add and modify in red part in our review. I hope you can understand. Thank you very much again. Answers to your comments were as follows:
Point 1: The authors list important articles in the manuscript, without their description, e.g.:
- a) “Relationship between sarcopenia and physical activity in older people: a systematic review and meta-analysis”
- b) “Health Outcomes of Sarcopenia: A Systematic Review and Meta-Analysis”
Response 1: Thanks for your comment. According to your comment, we have described the two important articles you proposed in the corresponding references of the review (reference 10:Health Outcomes of Sarcopenia: A Systematic Review and Meta-Analysis,line 68-69; reference 102: Relationship between sarcopenia and physical activity in older people: a systematic review and meta-analysis,line 312-315). In addition, we also rechecked all cited references and found there were seven references we cited but not described. We also described all these references in the corresponding references of the review (reference 3-7: line 57-63; reference 100: line 311-312; reference 101: line 311-312).
Point 2: The authors do not cite and describe important articles, e.g.
- a) “Interventions for Treating Sarcopenia: A Systematic Review and Meta-Analysis of Randomized Controlled Studies”
- b) „The Role of Muscle Mass Gain Following Protein Supplementation Plus Exercise Therapy in Older Adults with Sarcopenia and Frailty Risks: A Systematic Review and Meta-Regression Analysis of Randomized Trials”
- c) “Exercise interventions in healthy older adults with sarcopenia: A systematic review and meta-analysis”
- d) “Sarcopenia and its association with falls and fractures in older adults: A systematic review and meta-analysis”
- e) “Diagnosis, prevalence, and clinical impact of sarcopenia in COPD: a systematic review and meta-analysis”
The authors should review the link below and include the most important meta-analyzes in the manuscript. Authors should describe what distinguishes their article from other meta-analyzes published so far. This is the basis of my review. The text must contain missing articles (meta-analyzes) and a description of why your article is new. https://pubmed.ncbi.nlm.nih.gov/?term=sarcopenia%20meta-analysis
Response 2: Thanks for your comment. We opened the above link you sent to us and found there were 371 literatures on sarcopenia and meta-analysis. We reviewed all the literature and divided them into four important categories by research contents, including diagnostic criteria, prevalence, adverse health-related outcomes and interventions. However, due to the large number of meta-analyses on the same research contents, we can only cite some recently published important meta-analyses, because the recently published meta-analyses can basically cover all the present original studies in this field. Finally, we cited and described 13 additional important meta-analyses in our review. The detailed information about cited references were as follows:
As for the diagnostic criteria and prevalence of sarcopenia, references 3-9 (line 57-67) have been already cited and described. As for the adverse outcomes of sarcopenia, it mainly include three aspects: functional decline、disability、healthcare costs、falls、fractures、hospitalization、longer hospital stay, and mortality; poorer prognosis (overall survival, all-cause mortality, disease-free survival, cancer-specific survival, recurrence-free survival, postoperative complications) of many cancers; the occurrence of some diseases (dysphagia, cognitive impairment, hypertension, diabetes, osteoporosis, nonalcoholic fatty liver disease and depression). The first aspect has already been cited in reference 10-12 (line 67-69). So, we added the cited references for the second (reference 13, line 70-71) and third aspects (reference 14-20, line 72-75). As for your fourth meta-analysis (d) suggested to us, we reviewed it and found it indicated sarcopenia can lead to fractures and falls in older adults. Because the meta-analysis (reference 10: Health Outcomes of Sarcopenia: A Systematic Review and Meta-Analysis) we have cited also found sarcopenia can lead to fractures and falls in older adults, so we did not cite it repeatedly. As for interventions, it mainly included nutrition, exercise and combination. In addition to three meta-analyses (a, b and c) you suggested, we have also cited and described two additional meta-analyses. (Reference 108-112, line 399-404). In addition, disease-related associated factors are exposure factors in our inclusion criteria, while COPD in the fifth meta-analysis (e) you proposed is taken as the research participant. Because it is inconsistent with our research aim, so we did not cite it.
In response to your second question (Authors should describe what distinguishes their article from other meta-analyzes published so far and a description of why your article is new), we described it in the discussion part (line 256-262).
Point 3: The abstract should contain the most important results of the meta-analysis performed.
Response 3: Thanks for your comment. According to your comment, we have added the most important results of the meta-analysis performed. (See the red part in Abstract).
Point 4: I would suggest a more detailed description of the early identification of high-risk individuals and the development of relevant interventions to prevent sarcopenia in the elderly community. The authors have discussed this aspect too generally.
Response 4: Thanks for your comment. According to your comment, we have added the 4.4 Implications of The Review for Future Clinical Practice in the discussion part in order to a more detailed description of the early identification of high-risk individuals and the development of relevant interventions to prevent sarcopenia in the elderly community. (Discussion: 4.4 Implications of The Review for Future Clinical Practice; line 379-407).
Point 5: For the sake of clarity, I would sort the predictors in the tables according to their increasing importance (based on the obtained results).
Response 5: Thanks for your comment. According to your comment, we have sorted the predictors in the tables according to their increasing importance (based on the obtained meta-analyses results: the size of the OR value). (Table 1 and Table 2).

Reviewer 2 Report
This would be good to put the funnel plots and data showing publication bias in the present form, not requested. Typo at the line 111 should be fixed. Then it will be good to go. Well done!
The author reviewed enough number of studies and literatures through DB to perform systematic review and meta-analysis; to find out 'associated factors of sarcopenia in community-dwelling older adults'; three certain factors below, 1) sociodemographic, 2) behavioral, 3) disease-related , as title was. At one comments of the title, the study is closer to 'systematic review'rather than 'meta-analysis. In order of study selection, data extraction, and quality assurance at the Methods, there is no question, which is okay. With these study methods, the author described how studies were reviwed with PRISMA, which is an identical way to show CONSORT diagram at clinical trial. The analysis methods and results of data is well decribed with the Table 1 and 2. It would be better to provide odds ratio and funnel plots, not tables.
Author Response
Dear reviewer:
Thanks very much to you for giving us the opportunity to revise the review. According to your comments, we have made corresponding modifications in the review. Because the overall repetition rate of our review exceeded the 4% required by the journal, we tried to decrease repetition. In the process of decreasing repetition, we made a lot of small modifications while keeping the original contents unchanged. Using the function of track changes may make our review look particularly messy, so I marked the contents you suggested to add and modify in red part in our review. I hope you can understand. Thank you very much again. Answers to your comments were as follows:
Point 1: This would be good to put the funnel plots and data showing publication bias in the present form, not requested. Typo at the line 111 should be fixed. Then it will be good to go. Well done!
Response 1: Thanks for your comment. According to your comment, we have added the funnel plots and data (the number of missing studies and odds ratio) in the part 3.6 Publication Bias of our review. (See the red part, line 250-254; funnel plots in the supplemental material: Figure S33-S45). In addition, typo at the original line 111 (present line 127-128) has bene fixed.
Point 2: The author reviewed enough number of studies and literatures through DB to perform systematic review and meta-analysis; to find out 'associated factors of sarcopenia in community-dwelling older adults'; three certain factors below, 1) sociodemographic, 2) behavioral, 3) disease-related, as title was. At one comments of the title, the study is closer to 'systematic review' rather than 'meta-analysis. In order of study selection, data extraction, and quality assurance at the Methods, there is no question, which is okay. With these study methods, the author described how studies were reviewed with PRISMA, which is an identical way to show CONSORT diagram at clinical trial. The analysis methods and results of data is well described with the Table 1 and 2. It would be better to provide odds ratio and funnel plots, not tables.
Response 2: Thanks for your comment. Our review not only systematically reviewed the original studies on the factors associated with sarcopenia, but also provides a quantitative analysis of the factors associated with sarcopenia. Therefore, our view title should include systematic review and meta-analysis. According to your comment, we have added odds ratio and funnel plots in the part Publication Bias of our review (See the red part, line 250-254; funnel plots in the supplemental material: Figure S33-S45) and the results of publication bias (P value) were removed from the tables. In addition, we also added the forest plots from the meta-analysis in the supplemental material to show our meta-analysis process more clearly. (Figure S1-S32).

Round 2
Reviewer 1 Report
The authors corrected the manuscript with the indicated arguments.